# TRIM5α: A Protean Architect of Viral Recognition and Innate Immunity

**DOI:** 10.3390/v16070997

**Published:** 2024-06-21

**Authors:** Stephanie J. Spada, Michael E. Grigg, Fadila Bouamr, Sonja M. Best, Peijun Zhang

**Affiliations:** 1Division of Structural Biology, Wellcome Trust Centre for Human Genetics, University of Oxford, Oxford OX3 7BN, UK; peijun.zhang@strubi.ox.ac.uk; 2Laboratory of Parasitic Diseases, NIAID, NIH, Bethesda, MD 20894, USA; griggm@niaid.nih.gov (M.E.G.); bouamrf@niaid.nih.gov (F.B.); 3Laboratory of Neurological Infections and Immunity, Rocky Mountain Laboratories, NIAID, NIH, Hamilton, MT 59840, USA; sbest@niaid.nih.gov; 4Diamond Light Source, Harwell Science and Innovation Campus, Didcot OX11 0DE, UK; 5Chinese Academy of Medical Sciences Oxford Institute, University of Oxford, Oxford OX3 7BN, UK

**Keywords:** TRIM5α, retroviruses, orthoflaviviruses, orthopoxviruses, innate immunity, structure

## Abstract

The evolutionary pressures exerted by viral infections have led to the development of various cellular proteins with potent antiviral activities, some of which are known as antiviral restriction factors. TRIpartite Motif-containing protein 5 alpha (TRIM5α) is a well-studied restriction factor of retroviruses that exhibits virus- and host-species-specific functions in protecting against cross-primate transmission of specific lentiviruses. This specificity is achieved at the level of the host gene through positive selection predominantly within its C-terminal B30.2/PRYSPRY domain, which is responsible for the highly specific recognition of retroviral capsids. However, more recent work has challenged this paradigm, demonstrating TRIM5α as a restriction factor for retroelements as well as phylogenetically distinct viral families, acting similarly through the recognition of viral gene products via B30.2/PRYSPRY. This spectrum of antiviral activity raises questions regarding the genetic and structural plasticity of this protein as a mediator of the recognition of a potentially diverse array of viral molecular patterns. This review highlights the dynamic evolutionary footprint of the B30.2/PRYSPRY domain in response to retroviruses while exploring the guided ‘specificity’ conferred by the totality of TRIM5α’s additional domains that may account for its recently identified promiscuity.

## 1. Introduction

Restriction factors are diverse cell-intrinsic proteins encoded in the mammalian germline that have critical roles in cellular defense by inhibiting virus propagation [1]. In addition, some restriction factors act as pattern-recognition receptors (PRRs) upon sensing viral pathogen-associated molecular patterns (PAMPs) to trigger the activation of transcription factors and thereby control the expression of interferons (IFNs) and proinflammatory cytokines [1,2]. Type I, II, and III IFNs are responsible for establishing a cellular antiviral state by inducing the expression of IFN-stimulated genes (ISGs), many of which encode restriction factors [3,4,5]. These host cellular proteins function as barriers through many mechanisms, including the recognition of essential structurally conserved products encoded by viral genomes to interfere with specific stages of replication [5]. Alternatively, restriction factors may indirectly hinder viral accessibility to critical hijacked cellular resources [5]. IFNs upregulate hundreds of genes with antiviral activity against a broad spectrum of viruses. However, some of these genes are extremely important because they influence the outcome of a specific family of viral infections, including whether the virus can cross species barriers to establish transmission in new species. A major example of this is TRIpartite Motif-containing protein 5α (TRIM5α), which is the subject of this review.

Restriction factor genes that interface directly with viral pathogens are subject to evolutionary pressures amounting to a genetic arms race. Viruses adapt to evade these defenses, while hosts develop new strategies to counter viral antagonism. Ongoing evolution shapes the genetic landscape as both sides compete for survival. Thus, mammalian genomes demonstrate millions of years of positive selection, a key mechanism favoring the spread of advantageous genetic variants within a population [3]. Retroviral gene product integrations are believed to have stimulated the emergence of these defensive barriers. Fv1, which was derived from the retroviral *gag* gene, was among the first discovered in mice and functions as an active restriction factor against murine retroviruses [6]. The requirement for these structure–function relationships to be maintained results in the repeated exposure of the cell to conserved molecular patterns that help the cell to discriminate between self and non-self (or stressed self). The existence of conserved molecular patterns drives adaptive changes in host immunity genes, allowing the variability of multiple alleles in the gene pool to increase variant frequency, and such is the case for *Fv1* [7]. In addition to multiple alleles, these genes may also experience episodes of gene innovations, as is the case for many TRIpartite Motif (TRIM) proteins, including TRIM22 and TRIM5α [8]. For *TRIM5*, repeated pathogen conflict has led to an evolutionary record that represents adaptations to different viruses and has led to host-species-specific restriction (further detailed in Table 1).

As many as 100 proteins belong to the TRIM superfamily within the human genome, regulating functions in cell signaling and proliferation [18,19]. TRIM proteins with antiviral roles function in the modulation of cytokine signaling, the regulation of autophagy-mediated antiviral mechanisms, and direct inhibition by targeting specific viral proteins for degradation or otherwise disrupting their function [20]. TRIM proteins canonically consist of a RING domain, one or two B-box domains, and a coiled-coil domain, collectively called the RBCC motif (outlined in Figure 1a). The N-terminal RING finger domain is responsible for E3 ligase activity, such as ubiquitylation, ISGylation, and SUMOylation [21]. The B-box and coiled-coil domains work together to form higher-ordered TRIM assemblies [22,23,24,25]. TRIM5α, the longest splice variant of the *TRIM5* gene, contains a unique C-terminal B30.2/PRYSPRY (hereon referred to as SPRY) domain, is only present in vertebrates, and is generally accepted as a substrate recognition tool. The basic restriction mechanism involves the TRIM5α-mediated recognition and disruption of the viral core of incoming retroviruses shortly after fusion of the viral and cellular membranes [reviewed extensively in [26]. TRIM5α was originally identified as a potent inhibitor of early stages of Human Immunodeficiency Virus (HIV-1) replication within Old World monkey (OWM) cell lines [13]. Early studies suggested that, compared with OWM rhesus macaques, the human TRIM5α gene has a weaker affinity for cores from HIV-1 [13,27]. Today, it is recognized that viral escape from human TRIM5α-mediated restriction is an important feature of pandemic HIV-1 (subtype M) and that TRIM5α is likely a critical barrier to HIV-1 subtypes that have emerged in human populations but have not been widely transmitted, marking an interesting interspecies genomic divergence [28]. Positive selection in SPRY domain sequence variation among primate species is linked to target recognition and restriction efficiency [27,29]. Thus, the evolution of *TRIM5* is tied to the sequence diversity of the capsid protein (CA), of which ~1500 monomers assemble to form the mature viral core [30,31,32]. These discoveries were some of the first examples of host- and virus-species-specific restrictions. 

While the evolutionary conflict driving TRIM5α adaptation in primates appears to be due to repeated retroviral challenges, *TRIM5* has retained functional plasticity to restrict the replication of retroelements, such as LINE-1 ribonucleoprotein complexes, and unrelated viruses [33]. The discovery of the antiviral activity of TRIM5α against orthoflaviviruses challenged the understanding that the restriction potential was exclusively directed against retroviruses [34]. Through the SPRY domain, TRIM5α was shown to bind to the protease domain of nonstructural protein 3 (NS3), which, together with its virally encoded co-factor, NS2B, possesses enzymatic activity responsible for the cleavage of the viral polyprotein to liberate nonstructural proteins and initiate replication in orthoflaviviruses [34]. The trypsin-like serine protease is widely conserved among orthoflaviviruses and additionally cleaves cellular substrates to evade antiviral responses, including autophagy and innate immune signaling [35,36,37]. Finally, a recent study showed that TRIM5α can also restrict orthopoxviruses via the recognition of the highly conserved capsid protein L3 by the SPRY domain, thereby blocking the transcription of early viral genes [38]. Therefore, there is clearly more to uncover regarding the mechanisms of actions used by TRIM5α against medically important viruses. 

While there is a common requirement for the SPRY domain in the recognition of viral molecular patterns from all three viral families (retroviruses, orthoflaviviruses, and orthopoxviruses), how the positive selection of SPRY sequences and the higher-order assembly of TRIM5α influences the recognition of different viral families has not been explored. This review highlights the diversity of the *TRIM5* gene, describing episodes of genomic innovation and positive selection that optimize CA recognition. Additionally, we examine the current structures of TRIM5α while exploring the critical functionality of oligomerization and ubiquitination against CA that may also collectively exist to facilitate its recently discovered spectrum of restriction capabilities. 

## 2. TRIM5α in Antiviral Defense: CA Recognition, Immune Signaling, and Restriction

Viral evasion from TRIM5α and other proinflammatory signaling proteins is considered pivotal in determining the transmission potential of HIV-1 [28]. Following infection, intact viral cores are transported to the nuclear pore for the delivery of reverse-transcribed viral DNA to sites of provirus integration within the nucleus. TRIM5α’s function is to intercept incoming viral cores during cytoplasmic transport and disrupt reverse transcription [26]. The HIV-1 CA lattice serves as a template for complementary hexagonal arrays of TRIM5α that are believed to cage and disrupt mature cores [14,23,39,40,41,42]. Pathogen-mediated templating of TRIM5α activates its function as a PRR and immune sensor, helping to establish an antiviral state [40,43,44,45]. The latter involves the catalytic activity of the RING domain, which generates K63-linked ubiquitin chains [40,46]. These chains activate the ubiquitin-sensing kinase TGFβ-activated kinase 1 (TAK1), a key regulator of innate immunity and the proinflammatory signaling pathway [40]. Although questions remain regarding the precise mechanism by which the binding of TRIM5α results in core inactivation, research examining the contributions of individual domains to function has revealed a high degree of cooperation between the RING, B-box, coiled-coil, and SPRY domains. The following sections discuss these roles as they relate to the plasticity of virus recognition. 

## 3. SPRYing into Evolution

### 3.1. Retroviral Battles Shape Genomic Adaptations

Despite sharing 87% sequence identity with rhesus macaque TRIM5α, early studies show that the human TRIM5α ortholog only modestly restricts HIV-1 [27]. It is, however, a potent inhibitor of N-tropic murine leukemia viruses (N-MLV) and equine infectious anemia virus (EIAV) [12,16,47]. In contrast, New World monkey (NWM) cell lines are permissive to HIV-1, yet simian immunodeficiency virus (SIV) is blocked, supporting an established evolutionary record of host interspecies divergence where polymorphisms selected for intrinsic resistance to diverse retroviruses have become fixed [48]. 

High rates of non-synonymous substitutions per non-synonymous site (dN) to synonymous substitutions (dS) within *TRIM5* indicate accelerated protein evolution. It has been estimated that positive selection has shaped the gene for over 77 million years and that this evolutionary influence may be ongoing in many lineages [49]. For example, the cow genome includes five *TRIM5* genes, one of which, *TRIM5–3*, is active against HIV-1 and exhibits an abnormally high dN/dS rate, presumably sampling for better capsid-binding capabilities [50]. Similar events are seen within bats and rodent genomes that have an expansion of TRIM5-like orthologs, likely retained to facilitate independent evolution in response to infection with different viruses [3,8]. Evolutionary studies of *TRIM5* transcripts also show episodes of gene expansion, such as duplications in African green monkeys and even triplications in woolly and spider monkeys [29]. Positive selection is concentrated in the SPRY domain, specifically within four hypervariable variable loops determined from crystal structures that are linked to restriction efficiency. The crystal structure of rhesus macaque TRIM5α SPRY (PDB 4B3N) denotes a bent β-sandwich fold where conserved hydrophobic core residues are buried, with two antiparallel β-sheets consisting of tightly packed β-strands that maintain structural stability and an N-terminal α-helix [51] (Figure 1c). Moreover, gene expansion events paralleled the emergence of various endogenous retroviruses in OWMs, such as a 26-residue expansion in variable loop one (v1) coinciding with the introduction of different Human Endogenous Retroviruses (HERVs) and expansions in v3 of NWMs’ genomes coinciding with the emergence of HERV families [27]. Figure 1b highlights examples of genomic innovations within a region of the v1 loop between primates. 

TRIM5-mediated antiviral activity is robustly linked to the α isoform that contains the SPRY domain. It is also important to note that selective pressures resulted in a retrotransposition event that led to the insertion of cyclophilin A (CypA) on the C-terminal end of the TRIM5 gene in lieu of SPRY, as discovered in NW owl monkeys and independently in OWMs [52,53]. The emergence of the chimera, TRIM5CypA, renders owl monkey species greatly resistant to HIV-1 infection [53]. CypA demonstrates substantially higher sensitivity to the CA of HIV-1 (*K*D~10 μM) than rhesus macaque SPRY (estimated *K*D > 1 mM), suggesting that this affinity increase is a result of a selective advantage [54]. This interaction has also been demonstrated to be mediated by pattern-sensing, particularly with respect to the intrinsic curvature of CA geometry [55,56]. Interestingly, in human cells, HIV-1 recruits CypA to the capsid to disguise or coat itself to avoid recognition by TRIM5α, further suggesting a close evolutionary relationship between these two proteins [57]. 

### 3.2. The Structural Dynamics and Evolutionary Resilience of the SPRY Domain’s v1 Loop 

The SPRY domain displays structural divergence within four hypervariable loops, where almost half of the sequence variability occurs. The v1 loop, which is highly disordered and flexible, functions in conjunction with linker 2 (L2), which serves as the bridge between the coiled-coil and SPRY domains. V1 is therefore assumed to be incredibly mobile, extending from the rest of the domain to adopt various conformations, allowing interactions with viral epitopes of changing forms, thus contributing to the region’s structural plasticity [51]. 

Rhesus macaques retain multiple alleles of TRIM5α with divergent antiretroviral specificity when tested pairwise, largely linked to the presence of key amino acids on v1. Single point mutations within v1 of human TRIM5α have independently been shown to confer gain of restriction against HIV-1. Specific point mutations at positions like R332 and R335 have been shown to independently enhance restriction efficiency [58,59,60]. When combined, such as in the R332G-R335G pairing, they can even restore restriction to levels comparable to those seen in its rhesus macaque ortholog [59]. These positions belong to a positive selection cluster, previously reported as a ‘patch’ of residues between ~330 and 340, assumed to be a site of interaction with capsids [29] (Figure 1b). The v1 loop domain in human TRIM5α, particularly at critical mutation sites like R332, has been noted for its remarkable resilience to mutation. While natural sequences typically exhibit only three amino acids at this site, the introduction of a spectrum of mutations without a loss of function underscores the protein’s flexibility and tolerance [61]. This resilience not only facilitates adaptation but also brings human TRIM5α tantalizingly close to achieving potent functionality against HIV-1, often merely one mutation away. Uncovering mutations that empower human TRIM5α to effectively thwart HIV-1 could herald groundbreaking developments in gene therapy. Still, sampling single missense mutations might pose a considerable evolutionary challenge. Despite their rarity and high risk, indels (insertions or deletions) could potentially confer adaptive changes to overcome a viral challenge. Indels tend to concentrate in hypervariable regions and have been suggested to enable human TRIM5α to restrict simian lentivirus endemic to sabaeus monkeys (SIVsab) in a single evolutionary step, a feat that would otherwise necessitate five missense mutations [62]. These mutations, whether occurring alone or in conjunction, offer valuable insights into strategies for leveraging genetic modifications to enhance the protein’s ability to combat viral infections.

Recently, TRIM34 SPRY was found to function in tandem with the v1 loop of TRIM5α to limit viral cross-species transmission events [63]. Here, TRIM34, which has not undergone positive selection, utilizes the optimized CA-binding capabilities of TRIM5α to cooperatively function as an antiviral gene [63]. The TRIM6/34/5/22 gene cluster displays remarkably dissimilar patterns of TRIM gene accumulation in mammalian orders, potentially suggesting cooperation in antiviral action in a host-species-specific manner, thus contributing to functional plasticity in innate immunity [49]. 

## 4. Strategies of Molecular Interactions

TRIM5α’s limited affinity for individual CA monomers suggests that recognition is not solely determined by the SPRY sequence. Protein behavior is often governed by electrostatic effects, influencing molecular interactions, folding, and stability. Notably, the net electrostatic charge is a key determinant of HIV-1 and lentivirus restriction by TRIM5α [61]. Specifically, positive charges at positions 332 and 335 within the SPRY domain significantly impede human TRIM5α’s restriction of HIV-1. In fact, any nonpositive amino acid mutation at position 332 of human TRIM5α enhances HIV-1 restriction [61]. Furthermore, differences in electrostatic potential within the CA surface loop, targeted by rhesus macaque TRIM5α, explain the opposite restriction sensitivity of HIV-2 strain GH123 and SIVmac strain 239 [64]. The electrostatic potential is additionally reported to be important in facilitating coil-coiled interactions with autophagy factors [65]. 

Hydrophobic interactions also play a role in TRIM5α’s antiviral specificity, particularly evident in rhesus macaque TRIM5α. Notably, the rhesus macaque B-box domain exhibits an unusual property with two conserved hydrophobic patches among primate species [22]. Mutations within these patches dramatically reduce HIV-1 restriction, thereby contributing to overall antiviral potential [22]. Surface properties thus play a pivotal role in determining antiretroviral potential and may also offer insights into the molecular mechanisms of how TRIM5 recognizes novel viral targets. 

**Figure 1 viruses-16-00997-f001:**
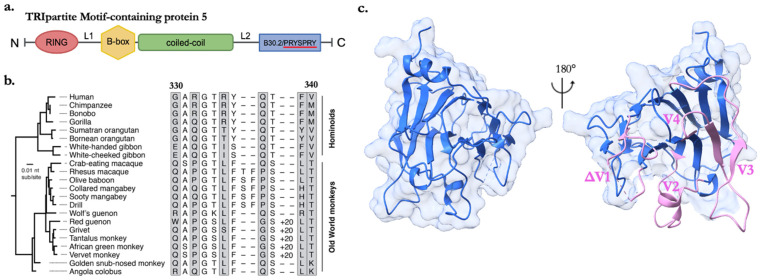
Insights into TRIM5α organization and sequence variability within the SPRY domain (**a**) Overview of canonical TRIM family RBCC motif with C-terminal SPRY domain. (**b**) Comparative analysis of TRIM5α sequences within the v1 loop (labeled 330–340, referencing human TRIM5α) from simian primates. The African green monkey clade contains a notable 20-amino acid duplication in the v1 loop, denoted as ‘+20′. Rapidly evolving residues are indicated with gray boxes. Reprinted from Ref. [61] under CC BY-NC 4.0. (**c**) The crystal structure of SPRY from the rhesus macaque (PDB: 4B3N). Variable loops 2–4 highlighted in pink (v2–v4). Missing v1 density depicted as ΔV1.

## 5. Exploring the Multivalency of the Coiled-Coil and B-Box Domains

Although the SPRY domain is absolutely required for substrate recognition, organized assembly mediated via the B-box and coiled-coil domains is critical in overcoming weak affinity for CA [22,66]. The coiled-coil domain exhibits an extended antiparallel dimer flanked by both the B-box and RING domains, which dimerizes in the low-order state and, cooperatively with B-box, multimerizes to stabilize a trimeric interface that allows the assembly of the hexagonal lattice around capsids [42,67,68]. The coiled-coil domain with L2 limits the flexibility of the SPRY loops, allowing more precise spacing to align with template epitopes, explaining the wide restriction potential for retroviruses that share common CA symmetry [69]. The coiled-coil domain also contributes to the spectrum of antiviral activity, with a few residues also under positive selection [29,70]. Sequence divergence analysis between squirrel monkeys and human TRIM5α reveals a subset of residues that are linked to the restriction potential of MLV [70]. Thus, the coiled-coil domain could function in tandem with the SPRY domain to strengthen interactions, potentially serving to strengthen direct interactions with the capsid in tangent with SPRY or indirect interactions to optimize the spatial arrangement of SPRY. Alternatively, the positive selection may be in response to maintaining additional interactions with host proteins or may even represent escape mutations from viral antagonists. 

The B-box domain has been reported to form both dimeric and trimeric interfaces that contribute to the oligomeric state of TRIM5α’s hexagonal net (Figure 2a–c). The B-box domain itself mediates flexible three-fold symmetric interactions at the vertices of the net that also allow perpendicular motions to facilitate curvature around CA, thus allowing the sampling and optimized binding of SPRY to target CA [68]. B-box interactions have also been suggested to contribute to the self-ubiquitination of TRIM5α as a PRR by promoting RING dimerization [40,46,71]. 

## 6. The RING Domain as an Effector of Restriction

TRIM5α drives proteasomal degradation and initiates innate immune signaling cascades upon capsid recognition [40,44,72]. Interestingly, the capsid is not the ubiquitination substrate for the E3 ubiquitin ligase RING domain; instead, TRIM5α undergoes autoubiquitylation, which is upregulated by CA binding [73,74]. Importantly, the ubiquitin–proteasome system is not strictly required, as adding proteasome inhibitors or introducing mutations within the RING domain does not restore infectivity, suggesting the presence of multiple barriers or blocks to infectivity [45,46,75,76]. 

The RING domain contributes to a cellular antiviral state via the production of K63-ubiquitin chains that can activate the TAK1 kinase complex promoting nuclear factor-κB (NF-κB) and/or activator protein 1 (AP-1) signaling pathways and induce the expression of cytokines (such as IFNβ, CXCL10, IL-6, and IL-8) [40,46,77]. K63-ubiquitin chain synthesis is driven by a catalytically active RING dimer that activates the ubiquitin-conjugating enzymes (E2s) UBE2W and heterodimeric UBE2N/V2, while a third RING domain promotes N-terminal autoubiquitylation [40,46,77] (Figure 3). The mechanism itself is driven by the asymmetry of the three RINGs brought into proximity via the B-box trimer in response to pattern recognition contacts between SPRY and CA [78] (Figure 3). This mechanism of N-terminal autoubiquitination is proposed to regulate innate immune signaling and protein turnover. It also supports the genetic adaptation of TRIM5α in combating retroviral CA proteins without requiring specific target protein sequences for ubiquitination.

Following its initial discovery, human TRIM5α was understood to promote HIV-1 restriction only weakly [10,13,16]. However, the cellular context under which this is examined is now known to be an important consideration. HIV-1 cloaks its core in CypA to hide from recognition by human TRIM5α, thereby reducing the impact of restriction [53,79,80]. In addition, restriction appears to be influenced by TRIM5 isoforms lacking a SPRY domain, and these variants, expressed at physiological levels, can act in a dominant-negative manner for the TRIM5α-mediated restriction of HIV-1 [13,81,82]. More recent studies have shown that the virus strain and cell type can influence human TRIM5α restriction mechanisms [81,83,84,85]. In fact, cytotoxic T lymphocyte escape CA mutants in individuals expressing HLA B27 or B57 alleles had increased sensitivity to the restriction of HIV-1 by human TRIM5α. This finding thus provides evidence that human TRIM5 can potently block infection and induce a cellular antiviral state [84,86,87]. Notably, human TRIM5α has also been found to contribute to the non-strain-specific suppression of HIV-1 in CD4+ T cells in a mechanism dependent on IFNα and the activation of the immunoproteasome [88]. Finally, polymorphisms in TRIM5 have been associated with the clinical course of HIV-1, suggesting a significant influence of this gene in humans [89]. 

## 7. Beyond Retroviruses: Novel Targets and Antiviral Mechanisms

While the evolutionary conflict driving TRIM5α adaptation in primates is repeated retroviral challenges, TRIM5α has retained remarkably functional plasticity to restrict the replication of unrelated viruses. Both human and rhesus macaque TRIM5α orthologs have been shown to recognize and degrade NS2B/NS3 from tick-borne orthoflaviviruses and are essential for the antiviral effects of type I IFN against sensitive orthoflaviviruses in human cells [34]. Orthoflaviviruses are single-stranded RNA viruses that replicate on modified membranes of the endoplasmic reticulum (ER) [90]. The orthoflavivirus genome encodes a single polyprotein that is post-translationally cleaved into at least three structural proteins and seven nonstructural proteins [90]. Key differences in the TRIM5α-mediated restriction of tick-borne orthoflaviviruses compared to retroviruses are that (I) a nonstructural protein with enzymatic function, NS3, is the target, and (II) TRIM5α must somehow access the replication organelle that is shielded by ER invaginations that prevent innate immune recognition of the replicating RNA to access NS3 [34]. In addition to tick-borne orthoflaviviruses, a second recent study showed human TRIM5α can also restrict DNA viruses, orthopoxviruses, for the recognition of the highly conserved capsid protein L3, blocking the transcription of early viral genes [38]. Like its role in suppressing retroviral replication, TRIM5α identifies and targets a structural protein located in the cytoplasm in the orthopoxvirus restriction model. However, these studies have also revealed that human TRIM5α induces the dimerization of L3 and triggers a post-translational modification suggestive of ubiquitination [38]. This modification occurs notably in the presence of wild-type TRIM5α, in contrast to a mutant version lacking E3 ubiquitin ligase activity [38].

TRIM5α demonstrates a common requirement for the C-terminal SPRY domain in the recognition of all three examples of viral molecular patterns, posing questions regarding how this domain has evolved the plasticity to recognize a broad range of viral structures. Alternatively, the viral proteins recognized by TRIM5α may have common structural features despite a lack of sequence identity. Moreover, findings from all three models suggest significant functions of the RING domain and oligomerization in TRIM5α’s activity. Dimerization contributes to the avidity of TRIM5α for the HIV-1 capsid and sets the stage for high-ordered assembly. Although it is unknown what role the multimerization of TRIM5α plays against orthoflaviviruses and orthopoxviruses, it is speculated that at least dimers of TRIM5α are involved in both [34,38]. 

It appears that the function of the TRIM5α E3 ubiquitin ligase domain can also be context-dependent. While the involvement of the ubiquitin–proteasome is variable in the TRIM5α-mediated restriction of HIV-1, the K48-dependent ubiquitin conjugation of the NS3-TRIM5α complex suggests that the proteasome-mediated degradation of NS3 is involved in the restriction of the tick-borne orthoflaviviruses [34]. The absence of increased reporter AP-1 gene expression when rhesus macaque TRIM5α was co-expressed with sensitive tick-borne orthoflaviviruses NS2B/NS3, along with unchanged cytokine expression (IFNβ, CXCL10, or IL8) upon TRIM5α depletion, suggests that TRIM5α may not function as a pattern-recognition receptor for the orthoflavivirus protease to induce cytokine expression [34]. An NF-κB reporter gene expression assay was also used to show that L3 expression could activate NF-κB independently of TRIM5α, and it was further suggested that activation may be independent of phosphorylated TAK1 [38]. It was, however, shown that wild-type TRIM5α boosted the stimulation of NF-κB activation, a feature lost in TRIM5α mutants lacking E3 ubiquitin ligase activity or their SPRY domain [38].

Additionally, it was reported that, unlike HIV-1, TRIM5-CypA fusion proteins do not restrict tick-borne orthoflavivirus replication, whereas, in HIV-1 and orthopoxviruses, the cellular protein CypA shields the mature core or capsid protein, allowing the evasion of this defensive barrier [34,38,57]. This may be a function of different viral requirements for CypA binding, as the requirement for CypA in orthoflavivirus replication is through the binding of two additional nonstructural proteins that are not targets of TRIM5α binding, thus further highlighting the importance of understanding the rules of structural recognition by TRIM5α [91,92]. For the first time, it has been demonstrated that a virally encoded protein can antagonize TRIM5α. The C6 protein, encoded by orthopoxviruses, binds to the RING domain of TRIM5α and induces its proteasomal degradation [38]. C6 is already known as a potent multifunctional inhibitor of interferon and, consequently, of ISGs [93]. These observations are summarized in Table 2, demonstrating how the genetic arms race continues to play out in surprising ways beyond the retroviruses. 

## 8. Conclusions

It has been suggested that the transmission potential of pandemic HIV-1 is linked to the evasion of TRIM5α and other proinflammatory signaling proteins, such as cGAS [28]. Therefore, a comprehensive understanding of cellular restriction factors and their mechanisms of action helps unravel the fundamental principles used by the cell to recognize non-self. TRIM5α, with its diverse orthologs and varying antiviral activity across species, exemplifies the ongoing genetic arms race between hosts and viruses, showcasing the intricate evolutionary dynamics underlying host–virus interactions. 

The pivotal role of the SPRY domain in recognizing viral molecular patterns underscores the importance of genomic innovation and positive selection in optimizing CA recognition. Additionally, the coordinated action of the RBCC motif facilitates TRIM5α’s multivalent interactions, leading to efficient viral restriction. Notably, the RING domain’s E3 ligase only promotes TRIM5α autoubiquitination to trigger innate immune signaling pathways, further enhancing antiviral defenses. Recent findings expand the repertoire of TRIM5α’s targets beyond retroviruses, demonstrating its ability to recognize and restrict orthoflaviviruses and orthopoxviruses. These discoveries highlight the dynamic nature of host–pathogen interactions and question how ongoing genetic and structural adaptations shape host defense mechanisms. The issue of whether the SPRY domain displays comparable selective genetic adaptations as observed in retroviruses, as well as the exact mechanisms by which E3 ligase activity and oligomerization contribute to TRIM5α’s varied role as an innate barrier, remains unresolved within the context of orthoflaviviruses and orthopoxviruses. Future structural determination of SPRY with its viral target and/or antagonist (as is the case for orthopoxviruses) will provide clues into the precise modes of recognition. Many TRIM proteins exhibit virus-specific restriction, with several TRIM proteins collectively contributing to effectively restrict a particular virus. At least six TRIM proteins have been shown to restrict the replication of retroviruses, some of which, such as TRIM25, act as co-factors [94]. Considering recent work demonstrating the cooperativity of TRIM5 and TRIM34 against HIV-1, future work might also reveal further observations that TRIM5α works in concert with other cellular factors to restrict this spectrum of viruses [63]. In addition, newly described roles for TRIM proteins in autophagy, immune signaling, cell differentiation, and apoptosis may hold clues into the vast antiviral potential observed. The TRIM family of proteins has been particularly closely linked to the induction of and relationship with autophagy factors [65,95]. In fact, some studies demonstrate the involvement of TRIM5α in the restriction of HIV-1 [95,96,97], thus broadening potential therapeutic targeting regimes.

Overall, unraveling the molecular mechanisms governing TRIM5α-mediated antiviral activity provides valuable insights into host immunity and may inform the development of novel therapeutic strategies to combat viral infections. Further exploration of TRIM5α’s evolutionary resilience and its diverse interactions with viral targets promises to uncover new avenues for understanding and harnessing innate immune responses against a broad range of viral pathogens.

## Figures and Tables

**Figure 2 viruses-16-00997-f002:**
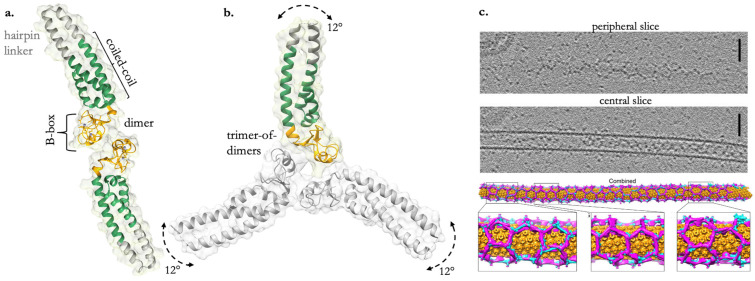
TRIM5α creates organized assemblies against CA. (**a**) The crystal structure of the BCC miniTRIM dimer (a biochemical and structurally friendly construct that preserves the integrated B-box/coiled-coil architecture of the full-length dimer) from PDB: 5EIU. (**b**) The BCC miniTRIM trimer from PDB: 5IEA. Arrows indicate coiled-coil domain shifts in relation to the B-box trimer, creating an arc of approximately 12° along this path. (**c**) Cryo-ET and subtomogram averaging of in vitro-assembled HIV-1 capsid tubes coated with hexagonal arrays of TRIM5α. Scale bars, 100 nm. A combined lattice map of TRIM5 dimers and trimers, with capsid hexamers colored in orange, TRIM dimers in cyan, and TRIM trimers in magenta. Reprinted from Ref. [42] under CC BY-NC 4.0.

**Figure 3 viruses-16-00997-f003:**
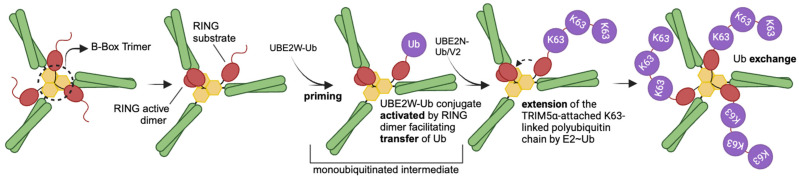
The autoubiquitylation of TRIM5α on retroviral capsids involves intricate interactions between RBCC domains [78]. The trimerization of B-box domains allows RING domains to be brought into proximity, forming an asymmetric structure that allows the RING domain to function as the catalytic activator and the substrate of ubiquitin. This RING catalytic activity initiates ubiquitination by activating an E2~Ub conjugate, UBE2W-Ub, and monoubiquitinates the N-terminus of the free RING. UBE2N-Ub/V2 then favors this monoubiquitinated site and promotes the subsequent extension and further exchange of K63-linked polyubiquitin chains. B-box (yellow); RING (red); coiled-coil (green); K63 ubiquitin (purple).

**Table 1 viruses-16-00997-t001:** TRIM5α orthologs possess a spectrum of antiviral activity that displays species-specific restriction against retroviruses [9,10,11,12,13,14,15,16,17]. “+” denotes restriction, “-” denotes no restriction, and “weak” denotes weak restriction. Abbreviations: SIVmac (Simian Immunodeficiency Virus of Macaques); SIVagm (Simian Immunodeficiency Virus of African Green Monkeys); N-tropic murine leukemia viruses (N-MLV); N-tropic murine leukemia viruses; equine infectious anemia virus (EIAV).

TRIM5α	HIV-1	SIVmac	SIVagm	N-MLV	B-MLV	EIAV
human	weak	-	-	+	-	+
rhesus macaque	+	-	+	weak	-	+

**Table 2 viruses-16-00997-t002:** A summary of the described spectrum of TRIM5α-mediated antiviral activity against HIV-1, orthoflaviviruses, and orthopoxviruses [26,34,38]. “+” Indicates the presence or positive involvement of the characteristic; “-” indicates the absence or negative involvement of the characteristic; “++” indicates a strong positive involvement; “+/-” indicates a variable or moderate involvement; “?” indicates uncertainty or unknown involvement.

TRIM5α Characteristics	HIV-1	Orthoflaviviruses	Orthopoxviruses
Viral target	Capsid lattice (structural)	Viral protease (nonstructural)	Capsid protein L3 (structural)
RING	+	+	+
SPRY/B30.2	+	+	+
Oligomerization	+	+	+
TRIM-CypA	+	-	?
Proteasome	+/-	+ degradative	?
Ubiquitination of target	-	+ (K48)	+ ?
Human TRIM5α	+/-	++	++
Innate immune signaling	+	-	+/- (with recognition of L3)
Antagonist(s)	CypA	?	CypA; C6

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
