# Peer review of "TRIM5α: A Protean Architect of Viral Recognition and Innate Immunity"

_viruses, 2024, doi:10.3390/v16070997_

Round 1

Reviewer 1 Report

Comments and Suggestions for Authors

This is a very well-written, informative review on TRIM5alpha, that incorporates recent advances, particularly the discovery that TRIM5alpha targets viruses belonging to very different families, including DNA viruses.

I have no major criticism. Below I am listing all my minor suggestions/comments/questions, in the order in which they appear in the text, not by order of importance.

1.       Abstract and introduction. The authors write that antiviral ISGs are collectively known as “restriction factors”. I believe that some virologists working on viruses other than retroviruses would object to that. This wording is somewhat retrovirus-specific.

2.       Lines 40-41, “a few of these genes are extremely important”. To the unfamiliar reader, the sentence may sound like a few ISGs restrict a large number of viruses, which of course is not exactly true, because of virus specificities: the majority of antiviral effectors / restriction factors discovered so far inhibit only one family of viruses.

3.       Table 1 needs some fixing. More space is needed horizontally so that to better separate virus names. Shouldn’t the legend be underneath the table? In the legend, why are you explaining the acronyms for some viruses but not others? Also in the legend, spaces are needed before parentheses.

4.       Lines 75-78, sentence starting with “TRIM5alpha, the longest…”: “substrate recognition tool” should apply to the PRYSPRY domain, not the whole protein (as the protein as a whole has additional functions); perhaps the sentence needs some fixing.

5.       Line 81, “pandemic” HIV-1. It’s the regular HIV-1 we are talking about, which of course is a pandemic, but why bring this precision here?

6.       Line 82, “Interestingly” usually means that an observation is somewhat surprising, whereas it is, of course, totally expected that human TRIM5alpha would not restrict (or poorly restrict) HIV-1.

7.       Regarding the “weak” restriction of HIV-1 by TRIM5alpha stated here and also at lines 135-136: at least two different teams have observed that HIV-1 capsids from HLA.B27+ or B57+ elite controllers are more efficiently restricted by human TRIM5alpha compared to HIV-1 capsids from normal progressors; and other teams have shown that rare mutations affecting TRIM5alpha functions predispose humans to HIV-1 disease progression; in other words, there is existing in vivo evidence that TRIM5alpha influences HIV-1 pathogenesis in humans, which seems to be of relevance for this review.

8.       Line 109, the sentence starting with “While there is…” seems incomplete.

9.       Line 120, “Viral evasion of TRIM5alpha”: maybe you meant “from” ?

10.   Paragraph lines 183-202: there seems to be some contradictions, as it is first said that only 3 aa variants can exist at position 132, while it is later said that most mutations at this position increase the restriction of HIV-1, and it is also said that this v1 loop is very tolerant to mutations. The ideas in this paragraph may need to be clarified. In addition, there is published evidence that similarly to R332, most mutations at R335 also increase restriction of HIV-1 by human TRIM5alpha, and combining mutations at the two positions (ex. R332G-R335G) has an even higher effect, which seems to support one of the points that the authors are trying to make, which is that multiple residues in v1 are responsible for the restriction of a given retrovirus.

11.   Line 221: a reference has not been formatted.

12.   Chapters 7 and 8, discussions about why TRIM5alpha is able to target very distinct viruses. A thought: is it a possibility that TRIM5alpha does not act alone, and that different cofactors are responsible for it being able to target different viruses?

Comments on the Quality of English Language

Fine

Author Response

Point to point responses to reviewers’ comments

Reviewer #1:

Comment #1:     

Abstract and introduction. The authors write that antiviral ISGs are collectively known as “restriction factors”. I believe that some virologists working on viruses other than retroviruses would object to that. This wording is somewhat retrovirus-specific. 

We appreciate the reviewer pointing out this distinction and have worked to incorporate a more appropriate wording in the introduction and abstract as follows:

Lines 11-13: The evolutionary pressures exerted by viral infections have led to the development of various cellular proteins with potent antiviral activities, some of which are known as antiviral restriction factors.

Lines 30-36: In addition, some restriction factors act as pattern-recognition receptors (PRRs) upon sensing viral pathogen-associated molecular patterns (PAMPs) to trigger activation of transcription factors and thereby control expression of interferons (IFN) and proinflammatory cytokines [1, 2]. Type I, II, and III IFNs are responsible for establishing a cellular antiviral state by inducing the expression IFN-stimulated genes (ISGs), many of which encode restriction factor [3-5].

Comment #2:

Lines 40-41, “a few of these genes are extremely important”. To the unfamiliar reader, the sentence may sound like a few ISGs restrict a large number of viruses, which of course is not exactly true, because of virus specificities: the majority of antiviral effectors / restriction factors discovered so far inhibit only one family of viruses. 

We understand this could be misleading, so have changed the sentence structure to reflect their influence on a specific family of viruses as follows:

Lines 40-43: IFNs upregulate hundreds of genes with antiviral activity against a broad spectrum of viruses. However, some of these genes are extremely important because they influence the outcome of a specific family of viral infection, including whether the virus can cross species barriers to establish transmission in new species.

Comment #3:

Table 1 needs some fixing. More space is needed horizontally so that to better separate virus names. Shouldn’t the legend be underneath the table? In the legend, why are you explaining the acronyms for some viruses but not others? Also in the legend, spaces are needed before parentheses.  

Table 1 seems to have been warped during formatting. We have edited it for clearer spacing and placed the legend underneath the table rather than on top. All virus acronyms are included now and spacing should be resolved. We appreciate the attention to detail you have taken to identify these mistakes.

Comment #4:

Lines 75-78, sentence starting with “TRIM5alpha, the longest…”: “substrate recognition tool” should apply to the PRYSPRY domain, not the whole protein (as the protein as a whole has additional functions); perhaps the sentence needs some fixing.

Lines 80-82: TRIM5α, the longest splice variant of the TRIM5 gene, contains a unique C-terminal B30.2/PRYSPRY (hereon referred to as SPRY) domain, which only present in vertebrates and is generally accepted as a substrate recognition tool.

Comment #5:

Line 81, “pandemic” HIV-1. It’s the regular HIV-1 we are talking about, which of course is a pandemic, but why bring this precision here? 

We have opted to remove pandemic at Line 87. The original distinction was made to consider subtypes of HIV-1 and zoonoses of SIV to humans that only subtype M of HIV-1 has become pandemic. We did however decide to include this distinction in Lines 88-92 to emphasize evasion of TRIM5α-mediated restriction is a critical determinant in transmission potential.

Lines 88-92: Today, it is recognized that viral escape from human TRIM5α-mediated restriction is an important feature of pandemic HIV-1 (subtype M), and that TRIM5α is likely a critical barrier to HIV-1 subtypes that have emerged in human populations but have not transmitted widely, marking an interesting interspecies genomic divergence [28].

Comment #6:

Line 82, “Interestingly” usually means that an observation is somewhat surprising, whereas it is, of course, totally expected that human TRIM5alpha would not restrict (or poorly restrict) HIV-1. 

Removed “Interestingly” from the text as this sentence does not warrant it.

Comment #7:

Regarding the “weak” restriction of HIV-1 by TRIM5alpha stated here and also at lines 135-136: at least two different teams have observed that HIV-1 capsids from HLA.B27+ or B57+ elite controllers are more efficiently restricted by human TRIM5alpha compared to HIV-1 capsids from normal progressors; and other teams have shown that rare mutations affecting TRIM5alpha functions predispose humans to HIV-1 disease progression; in other words, there is existing in vivo evidence that TRIM5alpha influences HIV-1 pathogenesis in humans, which seems to be of relevance for this review. 

This is an excellent point we have reworked to specify the caveats of this initial idea of “weak” restriction of HIV-1 by human TRIM5α throughout the text:

Lines 87-88: Compared with OWM rhesus macaques, early studies suggested the human TRIM5α gene has weaker affinity towards cores from HIV-1 [13, 27].

Lines 88-92: Today, it is recognized that viral escape from human TRIM5α-mediated restriction is an important feature of pandemic HIV-1 (subtype M), and that TRIM5α is likely a critical barrier to HIV-1 subtypes that have emerged in human populations but have not transmitted widely, marking an interesting interspecies genomic divergence [28].

Lines 143-144: Despite sharing 87% sequence identity with rhesus macaque TRIM5α, early studies show the human TRIM5α ortholog only modestly restricts HIV-1 [27].

Additionally, we have added another paragraph specifying some of these distinctions:

Lines 309-324: Following its initial discovery, human TRIM5α was understood to promote HIV-1 restriction only weakly [10, 13, 16]. However, the cellular context under which this is examined is now known to be an important consideration. HIV-1 cloaks its core in CypA to hide from recognition by human TRIM5α thereby reducing the impact of restriction [53, 79, 80]. In addition, restriction appears to be influenced by TRIM5 isoforms lacking a SPRY domain and these variants expressed at physiological levels can act dominant-negative to TRIM5α-mediated restriction of HIV-1 [13, 81, 82]. More recent studies have shown virus strain and cell type can influence human TRIM5α restriction mechanisms [81, 83-85]. In fact, cytotoxic T lymphocyte escape CA mutants in individuals expressing HLA B27 or B57 alleles, had increased sensitivity to restriction of HIV-1 by human TRIM5α. This finding thus provide evidence that human TRIM5 could potently block infection and induce a cellular antiviral state [84, 86, 87]. Notably, human TRIM5α has also been found to contribute to a non-strain specific suppression of HIV-1 in CD4+ T cells in a mechanism dependent on IFNα and activation of the immunoproteasome [88]. Finally, polymorphisms in TRIM5 have been associated with clinical course of HIV-1 suggesting a significant influence of this gene in humans [89].

Comment #8:

Line 109, the sentence starting with “While there is…” seems incomplete.

This sentence has been rectified, thanks for this catch.

Lines 114-117: While there is a common requirement for the SPRY domain in the recognition of viral molecular patterns from all three viral families (retroviruses, orthoflaviviruses, and orthopoxviruses), how positive selection of SPRY sequences and higher-order assembly of TRIM5α influences the recognition of different viral families has not been explored. 

Comment #9:

Line 120, “Viral evasion of TRIM5alpha”: maybe you meant “from”?

Lines 125-126: Viral evasion from TRIM5α and other proinflammatory signaling proteins is considered pivotal in determining transmission potential of HIV-1.

Comment #10:

Paragraph lines 183-202: there seems to be some contradictions, as it is first said that only 3 aa variants can exist at position 132, while it is later said that most mutations at this position increase the restriction of HIV-1, and it is also said that this v1 loop is very tolerant to mutations. The ideas in this paragraph may need to be clarified. In addition, there is published evidence that similarly to R332, most mutations at R335 also increase restriction of HIV-1 by human TRIM5alpha, and combining mutations at the two positions (ex. R332G-R335G) has an even higher effect, which seems to support one of the points that the authors are trying to make, which is that multiple residues in v1 are responsible for the restriction of a given retrovirus. 

We appreciate this could have been a bit misleading the way it was previously written. Although only three amino acids are seen in natural sequences, Tenthorey et al. 2020 demonstrated that any mutation at position 332 that reduces positive charge increases restriction potential of human TRIM5α against HIV-1. This study highlights the evolutionary resilience of the v1 to tolerate missense mutations which could be a strategy evolved to optimize capsid binding. We have also edited this paragraph to include the observations that position 335 is additional an important determinant in CA binding.

Lines 192-215: Rhesus macaques retain multiple alleles of TRIM5α with divergent antiretroviral specificity when tested pairwise, largely linked to the presence of key amino acids on v1. Single point mutations within v1 of human TRIM5α have independently been shown to confer gain of restriction against HIV-1. Specific point mutations at positions like R332 and R335 have been shown to independently enhance restriction efficiency [58-60]. When combined, such as in the R332G-R335G pairing, they can even restore restriction to levels comparable to those seen in its rhesus macaque orthologue [59]. These positions belong to a positive selection cluster, previously reported as a ‘patch’ of residues between ~330-340 assumed to be a site of interaction with capsids [29] (Figure 1b). The v1 loop domain in human TRIM5α, particularly at critical mutation sites like R332, has been noted for its remarkable resilience to mutation. While natural sequences typically exhibit only three amino acids at this site, introduction of a spectrum of mutations without loss of function underscore the protein's flexibility and tolerance [61]. This resilience not only facilitates adaptation but also brings human TRIM5α tantalizingly close to achieving potent functionality against HIV-1, often merely one mutation away. Uncovering mutations that empower human TRIM5α to effectively thwart HIV-1 could herald groundbreaking developments in gene therapy. Still, sampling single missense mutations might pose a considerable evolutionary challenge. Despite their rarity and high risk, indels (insertions or deletions) could potentially confer adaptive changes to overcome viral challenge. Indels tend to concentrate in hypervariable regions and have been suggested to enable human TRIM5α to restrict Simian lentivirus endemic to sabaeus monkeys (SIVsab) in a single evolutionary step, a feat that would otherwise necessitate five missense mutations [62]. These mutations, whether occurring alone or in conjunction, offer valuable insights into strategies for leveraging genetic modifications to enhance the protein's ability to combat viral infections.

Comment #11:

Line 221: a reference has not been formatted.

Resolved. 

Comment #12:

Chapters 7 and 8, discussions about why TRIM5alpha is able to target very distinct viruses. A thought: is it a possibility that TRIM5alpha does not act alone, and that different cofactors are responsible for it being able to target different viruses? 

It is very possible TRIM5α functions in conjunction with other TRIM protein or cofactors to facilitate the observed broad-spectrum recognition and restriction potential descried in this review. This reflects a very interesting discussion into the mechanisms underlying innate immune recognition and synergy. Lines 216-222 reflect this concept in reference to the v1 loop of TRIM5α acting with TRIM34 to exert antiviral potential. We additionally reference a paper suggesting TRIM6/34/5/22 gene cluster interesting patterns of TRIM gene accumulation.

We have also added into the conclusions a bit of a discussion about cooperatively of TRIMs and the discussion of co-factors.

Lines 414-418: At least six TRIM proteins have been shown to restrict replication of retroviruses, some of which, such as TRIM25, acting as co-factors [94]. Considering recent work demonstrating a cooperatively of TRIM5 and TRIM34 against HIV-1, future work might also reveal and further observations that TRIM5α works in concert with other cellular factors to restrict this spectrum of viruses [63].

This sentence seems to be a repeat of the last sentence and can be removed

Reviewer 2 Report

Comments and Suggestions for Authors

TRIM5a is a well-known retroviral restriction factor. The SPRY domain of TRIM5a is critical to mediate recognition and binding of retroviral capsids. The manuscript aims to describe the role of TRIM5a in viral recognition and innate immunity. In general, the authors comprehensively reviewed most related papers to emphasize the importance of TRIM5a, especially in the part of the SPRY domain.  

1. The authors did not mention too much about innate immunity, but the title and the context mentioned the immune signaling (line 118).  

2. In Figure 3, the authors did not explain clearly how TRIM5a autoubiquitination is activated by capsid binding. And, the figure did not display the color shown represents which domain in TRIM5. 

3. Table 1, cite the reference for each species-specific antiviral activity by TRIM5a. The space needs to be enlarged between each species-specific restriction for publication.

Author Response

Point to point responses to reviewers’ comments

Reviewer #2:

Comment #1: The authors did not mention too much about innate immunity, but the title and the context mentioned the immune signaling (line 118).

The manuscript has been edited to include more information regarding basics of TRIM5α-mediated innate immune signaling with regards to HIV-1 restriction in section 2 and more extensively in section 6 (lines 290-301). Additionally, section 7 details what is known thus far about immune signaling against TRIM5α-mediated restriction of poxviruses and flaviviruses (lines 357-370).

Comment #2. In Figure 3, the authors did not explain clearly how TRIM5a autoubiquitination is activated by capsid binding. And, the figure did not display the color shown represents which domain in TRIM5.  

Figure 3 and the legend has been detailed for clarity and the mechanism has been rewritten in section 6 (lines 283-301). Colors labels were additionally added which match those of the TRIM5α domain organization in Figure 1a.

Comment #3:

Table 1, cite the reference for each species-specific antiviral activity by TRIM5a. The space needs to be enlarged between each species-specific restriction for publication. 

Table 1 seems to have been warped during formatting. We have edited it for clearer spacing and placed the legend under the table rather than on top. All virus acronyms are included now and spacing should be resolved. The citations detailed the antiviral specificities have been added as well.

Reviewer 3 Report

Comments and Suggestions for Authors

edits:

Please add some figures/ charts/tables explaining the actions of TRIM5a on various host and viral proteins. 

Table 1: Make the table more organized by highlighting the borders of rows and columns. 

line 70: rewrite the sentence for better understanding 

121: change the word transmission with better-suited words. 

131: "Plasticity of virus recognition" -What does this mean? please explain..

Author Response

Point to point responses to reviewers’ comments

Reviewer #3:

Comment #1:

Please add some figures/ charts/tables explaining the actions of TRIM5a on various host and viral proteins. 

We decided a table summarizing the range of antiviral activity across the three viral families described would be a nice addition based on your suggestion. Please see Table 2 added which summarizes the roles of TRIM5α in all three antiviral mechanisms described.

Comment #2:

Table 1: Make the table more organized by highlighting the borders of rows and columns.  

Table 1 needed some formatting readjustments. The citations detailed the antiviral specificities have been added as well.

Comment #3:

line 70: rewrite the sentence for better understanding

Lines 80-82: TRIM5α, the longest splice variant of the TRIM5 gene, contains a unique C-terminal B30.2/PRYSPRY (hereon referred to as SPRY) domain, which only present in vertebrates and is generally accepted as a substrate recognition tool.

Here, we are explaining that TRIM5α contains the C-terminal SPRY domain which is only present in vertebrates and is understood as the substrate recognition tool.

We believe the reviewer is responding to this sentence in our revised version. If we are not correct, we are happy to make further edits.

Comment #4:

121: change the word transmission with better-suited words. 

We have edited section 2 to look for clarity. However, are unsure if the reviewer is referencing lines 125-126 in reference to the word “transmission”. We are happy to make this edit if needed.  

Comment #5

131: "Plasticity of virus recognition" -What does this mean? please explain..  

The plasticity of virus recognition refers to how an individual protein (TRIM5α) with seemingly an evolutionarily record directed specifically to retroviruses, retains the ability to restrict unrelated viruses. The next few sections discuss how all the domains are organized and function with specificity to retroviral capsids.
